# Clinical Performance of MAGLUMI Diagnostic Tests for the Automated Detection of Dengue Virus

**DOI:** 10.3390/v17010106

**Published:** 2025-01-14

**Authors:** Bo Peng, Zhonggang Fang, Cong Li, Kun Liu, Ting Wang, Ke Huang, Fan Yang, Yalan Huang, Chunli Wu, Yue Li, Dana Huang, Qian Zhang, Yijun Tang, Xiaolian Liu, Wei Rao, Xiaolu Shi

**Affiliations:** 1Microbiology Laboratory, Shenzhen Center for Disease Control and Prevention, Shenzhen 518055, China; 2Key-Laboratory of Intelligent Tracking and Forecasting for Infectious Disease, NHC Key Laboratory of Medical Virology and Viral Diseases, National Institute for Viral Disease Control and Prevention, Chinese Center for Disease Control and Prevention, Beijing 102206, China; 3Research & Development Department, Shenzhen New Industries Biomedical Engineering Co., Ltd., No. 23, Jinxiu East Road, Pingshan District, Shenzhen 518122, China

**Keywords:** dengue virus, NS1, antigen, chemiluminescence immunoassay

## Abstract

Aims: The screening and diagnosis of dengue virus infection play a crucial role in controlling the epidemic of dengue fever, highlighting the urgent need for a highly sensitive, simple, and rapid laboratory testing method. This study aims to assess the clinical performance of MAGLUMI Denv NS1 in detecting dengue virus NS1 antigen. Methods: A retrospective study was conducted to assess the sensitivity and specificity of MAGLUMI Denv NS1 using residual samples. Dengue-confirmed and excluded samples, validated by qPCR, were subjected to testing with MAGLUMI Denv NS1 in accordance with the manufacturer’s instructions. The linear range, endogenous interference, and cross-reactivity of MAGLUMI Denv NS1 were verified, and a consistency analysis with commercial comparator products was carried out. Results: The diagnostic specificity of MAGLUMI Denv NS1 is 98.41% (62/63), and the sensitivity is 98.32% (117/119). It effectively detects various serotypes of dengue virus, with no observed endogenous interference or cross-reactivity. Additionally, the consistency of NS1, IgM, and IgG tests on the MAGLUMI platform compared to commercial comparator reagents reaches 85.71%, 99.25%, and 98.97%, respectively. Conclusions: The MAGLUMI Denv NS1 represents a highly sensitive laboratory testing method capable of enhancing the diagnostic accuracy and efficiency of dengue virus infection detection.

## 1. Introduction

Dengue virus infection is a prevalent mosquito-borne disease in tropical and subtropical regions. Globally, an estimated 390 million individuals are affected by dengue virus annually, resulting in over 500,000 hospitalizations and 25,000 fatalities [1]. Dengue virus can be categorized into four serotypes based on antigenic disparities known as DENV1-4 [2]. Infection with any of the four serotypes of dengue fever virus may lead to a spectrum of clinical outcomes ranging from mild fever to classic dengue hemorrhagic fever (DHF) and dengue shock syndrome (DSS) [3]. Although the genomes of these four serotypes are similar, they are unable to individually elicit cross-neutralizing antibodies. Severe dengue fever primarily occurs in individuals who have experienced secondary infections with different serotypes of the virus [4]. This is due to pre-existing non-neutralizing or low-affinity antibodies, which, upon encountering a heterologous serotype of dengue virus, are unable to effectively neutralize the new serotype and instead facilitate viral entry into host cells, leading to increased viral replication and inflammation response. Consequently, this increases the risk of developing severe dengue fever upon secondary infection—a phenomenon known as antibody-dependent enhancement (ADE) [5,6]. The intricate cross-reactivity among dengue serotypes poses a significant challenge in the development of effective vaccines for dengue fever. Hence, timely detection, accurate diagnosis, and comprehensive early intervention are pivotal for successful management of dengue fever [7,8].

The WHO guidelines recommended multiple diagnostic methods for dengue fever, including virus isolation, nucleic acid detection, antigen detection, serological testing, and hematological testing [9]. Despite being considered the most sensitive diagnostic approaches, virus isolation through cell culture and nucleic acid detection via real-time quantitative PCR have limitations in terms of rapid diagnosis due to their requirement for specialized knowledge, advanced facilities, expensive laboratory equipment, and a significant amount of time for virus isolation and RNA purification [10,11,12]. The non-structural protein 1 (NS1) is a highly conserved glycoprotein encoded by the dengue virus, which typically reaches high concentrations in the serum of dengue fever patients within 1–5 days after infection and is detected almost simultaneously with viral RNA, long before the production of antibodies. Therefore, NS1 antigen detection serves as an early diagnostic tool for dengue fever [13,14]. Furthermore, the presence of NS1 antigen for up to 9 days or even longer can indicate active viral replication and potential severity of the disease [15,16]. Additionally, a positive NS1 test result can differentiate from other causes of febrile illnesses, such as influenza, Zika virus infection, etc., thus enhancing diagnostic specificity [17]. At present, commercial products for the detection of dengue virus NS1 primarily consist of immunochromatographic assays for rapid on-site screening and enzyme-linked immunosorbent assays (ELISA) commonly utilized for laboratory diagnosis. Several studies evaluating the clinical performance of NS1 detection kits have reported a range of sensitivities from 48.5% to 80.7% and specificities from 92.5% to 99.4% for NS1 antigen rapid diagnostic tests [18,19,20,21], while the sensitivity of ELISA detection methods varies from less than 75% to as high as 100% [22,23]. A recent study on detection sensitivity indicated that combined testing of NS1/IgG/IgM (90.65%) demonstrated higher clinical performance compared to IgG (90.06%) or NS1 (87.50%) alone [24].

In order to enhance the sensitivity, stability, testing speed, and automation level of NS1 detection methods and address clinical rapid diagnostic needs and laboratory efficiency, we have developed a MAGLUMI Denv NS1 chemiluminescent immunoassay (CLIA). This assay employs the sandwich principle of dual antibodies to quantitatively detect dengue fever virus NS1 antigen in human serum and plasma for adjunctive diagnosis of dengue fever infection. In this investigation, we assessed the clinical performance of this assay.

## 2. Material and Methods

### 2.1. Clinical Samples

In this study, a total of 182 serum samples were collected from patients that were suspected to have dengue fever. Following real-time qPCR detection, 63 cases were excluded, and 119 were confirmed with a Ct value greater than 40 as the negative criterion. Information about the dengue-positive sample is presented in Appendix A.

All clinical samples underwent testing at the Center for Disease Control and Prevention of Shenzhen, Guangdong Province, China, between January 2019 and May 2023, with relevant clinical symptoms and demographic data being recorded (Figure 1).

### 2.2. MAGLUMI Dengue Virus NS1 Antigen Test

The MAGLUMI dengue virus NS1 antigen test utilizes a sandwich chemiluminescence immunoassay, in which the sample is fully mixed with magnetic microbeads coated with NS1 monoclonal antibody, N-(4-Aminobutyl)-n-ethylisoluminol (ABEI) labeled by another NS1 monoclonal antibody, and buffer solution to form a sandwich complex during incubation. The information of the antibodies independently produced by the company is shown in the following table:

**Coating Antibody****Labeling Antibody**ImmunogenDengue NS1 antigenDengue NS1 antigenSourcemousemouseMolecular weight (kDa)150150SDS-PAGE bands (kDa)25, 5525, 55Purity (SDS-page)≥95%≥95%Concentration (mg/mL)2.449–3.1782.114–2.869

Following precipitation in the magnetic field, the reaction liquid is introduced to initiate the chemiluminescence reaction. The resulting light signal is measured by photomultiplier tubes as a relative light unit (rlu), providing a proportional indication of the concentration of dengue virus NS1 antigen present in the sample.

In the same instrument system, Maglumi DENV IgG employs the Indirect Chemiluminescence Immunoassay. The prediluted sample, buffer, and magnetic microbeads coated with dengue virus antigen are mixed and precipitated in a magnetic field. Subsequently, ABEI labeled with anti-human IgG monoclonal antibody is added to form immuno-complexes. MAGLUMI DENV IgM utilizes the capture chemiluminescence immunoassay. The sample, buffer, and magnetic microbeads coated with anti-human IgM antibody are mixed and precipitated. Then, ABEI labeled with dengue virus antigen is added and incubated, reacting to form immuno-complexes.

### 2.3. Analytical Performance Verification

Three distinct batches of reagents were employed to test 4 blank serum samples, 4 low-concentration serum samples, and 6 low-concentration serum samples of known concentrations. LoB, LoD, and LoQ were evaluated, respectively. Each sample was tested over three days, with 5 replicates per day. Linearity verification was conducted in accordance with CLSI standards. The 6 samples provided by the manufacturer, each with a specified concentration, were tested three times within a single run. The mean value of the test results was calculated and then fitted linearly with the theoretical concentration using the least square method. Subsequently, the linear correlation coefficient was determined.

### 2.4. Endogenous Interference and Cross-Reactivity

A total of 15 samples containing potential cross-reactive substances and 37 samples with interfering substances were assessed. The former consisted of inactivated clinical samples or negative clinical samples supplemented with inactivated pathogen cultures, such as Zika virus, hepatitis A virus, hepatitis B virus, hepatitis C virus, and chikungunya virus. The sample matrix was consistent with the anticipated type of test sample. The latter comprised clinically negative samples supplemented with potential endogenous and exogenous interfering substances, including hemoglobin, HAMA, lipid emulsion, ANA, bilirubin, rheumatoid factor, acetaminophen, lysine acetylsalicylate (LAS), ibuprofen, and dexamethasone.

### 2.5. Statistical Analysis

Statistical analysis was conducted using MedCalc software version 18.11.3 and Microsoft Office Excel 2013. A four-grid table was established for qualitative data analysis, sensitivity and specificity were calculated as percentages, the Wilson score interval method was employed to calculate the 95% confidence interval of evaluation indicators, and a Kappa consistency test analysis was performed. Between-group comparison utilized the Wilcox rank sum test, with *p* < 0.05 indicating statistically significant differences. Furthermore, Graphpad Prism software was used to compare various test results and evaluate data correlation generated by two testing methods through simple linear regression analysis.

### 2.6. Ethics

The performance validation study was carried out by the Shenzhen Center for Disease Control and Prevention in accordance with the guidelines of the Helsinki Declaration and following the principles of good clinical practice. All samples used in this study were residual samples obtained with general ethical approval, and subjects provided broad informed consent through signature.

## 3. Results

### 3.1. Diagnostic Specificity and Sensitivity

The results of clinical sample testing using the MAGLUMI Denv NS1 test demonstrated that among 119 dengue-confirmed samples, 117 exhibited positive NS1 antigen reactivity, while only one out of 63 dengue-excluded samples showed NS1 antigen reactivity. Statistical analysis revealed a diagnostic sensitivity of 98.32% (117/119, 95% CI: 98–99%) and a diagnostic specificity of 98.41% (62/63, 95% CI: 98–100%) for the MAGLUMI Denv NS1 test (Table 1 and Table 2).

By categorizing 119 confirmed dengue samples based on serotypes, we can also determine the diagnostic sensitivity of the MAGLUMI Denv NS1 test for four distinct dengue virus serotypes. The aforementioned data illustrates that the MAGLUMI Denv NS1 test exhibits high diagnostic sensitivity and specificity and possesses the capability to detect various serotypes of the dengue virus (Table 3).

Through the assessment of data derived from a MAGLUMI Denv NS1 test and real-time qPCR utilizing simple linear regression analysis, it was determined that there exists no significant correlation between the two detection methodologies, potentially attributed to variances in detection principles (Appendix A).

### 3.2. Analytical Specificity

The analytical specificity of the MAGLUMI Denv NS1 test was determined to be 100% (52/52, 95% CI: 93.12–100%), as evidenced by the lack of reactivity in a total of 52 clinical samples that had been deliberately spiked with cross-interfering substances or a negative serum matrix containing such substances (Table 4).

### 3.3. Analytic Sensitivity and Linearity Verification

LoB, LoD, and LoQ were calculated based on the CLSI Guideline EP17-A2. The LOB for the MAGLUMI Denv NS1 test was 0.500 AU/mL, with the LoD and LoQ at 0.800 AU/mL and 1.000 AU/mL, respectively (Table 5).

The linearity verification results for the MAGLUMI Denv NS1 test are depicted in Figure 2. The best-fit linear regression equation is y=0.9196x−0.4553, with an R-squared value of 0.9978, indicating a significant correlation between the average measured values from three replicate experiments and the expected NS1 concentration within the range of 1~400 AU/mL.

### 3.4. Serum to Plasma Equivalency

Repeated tests were conducted on eight groups of negative serum or plasma samples treated with various substrates, all yielding negative results. Similarly, 11 groups of positive serum or plasma samples treated with different substrates were retested, and all showed positive results. Despite variations in AU/mL levels among different matrices, the determination of sample positivity or negativity remained unaffected (Table 6).

### 3.5. Consistency Between Different Dengue Diagnostic Assays

In order to assess the consistency of the MAGLUMI Denv NS1 test in clinical performance with commercial comparator reagents, a total of seven DEN-positive samples and 107 DEN-negative samples were subjected to testing using the MAGLUMI Denv NS1 test and EUROIMMUN Denv NS1, respectively. The results revealed a negative concordance rate of 99.07% between the two products. While the number of positive samples was limited, it was observed that the positive concordance between the two products could reach as high as 85.71% (Table 7).

Simultaneously, we conducted an evaluation of the newly developed dengue virus IgG and IgM reagent kits on the MAGLUMI Diagnostic Platform to assess their clinical performance consistency with comparable products. Our findings revealed a remarkable overall agreement rate of 98.97% for IgG and 99.25% for IgM when compared with EUROIMMUN DENV, indicating a high level of concordance between the dengue diagnostic kits based on the MAGLUMI Diagnostic Platform and recognized testing products (Table 6). These results support the utility of these kits for the simultaneous detection of three biomarkers at different stages of dengue infection.

## 4. Discussion

Commercial detection products with high specificity and sensitivity are crucial for the prompt diagnosis of dengue fever infection and epidemic prevention. The early presence and abundant levels of NS1 antigen in patient serum have led to its widespread utilization in various immunological analysis products based on monoclonal or polyclonal antibodies, establishing it as a well-researched biomarker for detecting dengue virus. This study presents the clinical performance of MAGLUMI Denv NS1 utilizing the fully automated chemiluminescence immunoassay platform MAGLUMI series. The results demonstrate that MAGLUMI Denv NS1 exhibits a sensitivity of 98.32% and a specificity of 98.41% for detecting dengue virus infection, surpassing the reported sensitivity of similar NS1 antigen detection products at 80.7%. These findings indicate that the exceptional clinical performance of MAGLUMI Denv NS1 meets the diagnostic requirements for dengue virus infection.

Although specific antiviral drugs for the treatment of dengue fever are currently unavailable, understanding the serotype of the virus that patients are infected with may assist clinical doctors’ assessment of disease severity [25,26]. The predominant virus infections in most dengue fever endemic areas globally are DENV-1 and DENV-2 serotypes, with DENV-2 being considered the most common serotype causing severe dengue fever [27,28]. Our study identified 99 samples positive for DENV-1 and 14 samples positive for DENV-2, with a high sensitivity of 112/113. Regrettably, we did not have sufficient positive samples of DENV-3 and DENV-4 serotypes for testing. Given that all four serotypes of dengue virus spread in a similar manner, prior to the application of MAGLUMI Denv NS1 in dengue-endemic areas, verification will be carried out on a larger sample size, especially for samples of DENV 3 and DENV 4.

Cross-reactivity is a significant factor impacting the precision of dengue virus diagnosis. The clinical presentations of dengue fever and other arthropod-borne viral diseases, such as chikungunya fever and Zika fever, can be quite similar at times, manifesting with symptoms like fever, rash, and joint pain [29,30]. This similarity complicates the diagnostic process, necessitating comprehensive laboratory testing and epidemiological history for accurate diagnosis. While our study had a limited sample size, we did not observe cross-reactivity of MAGLUMI Denv NS1 with Zika virus or chikungunya virus. Additionally, we confirmed the excellent detection performance of MAGLUMI Denv NS1 for endogenous interfering substances and its suitability for testing serum or plasma samples from various matrices.

The dengue fever IgM antibody is produced in the early stages of infection, typically beginning to rise 3–7 days after infection and gradually decreasing over time. A positive IgM result indicates a recent or ongoing dengue virus infection, which is crucial for distinguishing between recent and past infections. In contrast, the dengue fever IgG antibody develops during the later phase of infection and throughout the recovery period, persisting for an extended duration [31,32]. It not only signifies a history of prior dengue virus infection but also serves as a tool for assessing potential enhancement effects (ADE) in cases of secondary infections, as a subsequent infection may lead to more severe manifestations of dengue fever. The concurrent detection of dengue virus NS1 antigen, IgG antibodies, and IgM antibodies has the potential to enhance the sensitivity and specificity of diagnosis [33,34]. On the other hand, the antibodies produced by the primary dengue infection might compete with the reagents, thereby diminishing the performance of the detection during the secondary infection. Among the 119 samples of dengue-infected patients employed in this study, seven cases were IgG positive and were within the 4th to 7th days of fever, indicating that they could be patients with secondary infections, and all were detected by the MAGLUMI Denv NS1 test (Appendix A). To further validate the analytical specificity of the reagent, in future studies, it is necessary to test more acute specimens and compare the performance of NS1 in IgG-positive and IgG-negative samples.

In our study, the overall agreement rate between IgG and EUROIMMUN DENV IgG was 98.97%, while the overall agreement rate between IgM and EUROIMMUN DENV IgM reached 99.25%. These findings demonstrate that the dengue diagnostic kit based on the MAGLUMI Diagnostic Platform exhibits a high level of concordance with recognized similar testing products, thereby enabling its utility for the combined detection of three markers at various stages of dengue infection. Regrettably, limited availability of dengue-positive samples precluded separate investigation into dynamic changes in levels of dengue virus markers during primary and secondary infections. To facilitate improved assessment by healthcare professionals regarding the stage of dengue infection and to guide subsequent treatment plans, further validation is warranted to confirm quantitative detection performance of MAGLUMI Denv NS1 as well as IgG and IgM across different stages of infection.

MAGLUMI Anti-NS1 represents a direct chemiluminescence immunoassay (DCLIA) approach, featuring simplistic technology and a fully automated operational procedure. The time to results merely demands 37 min, which is significantly shorter than the approximately 2 h requisite for the current ELISA reagents [23]. Simultaneously, the diagnostic performance of MAGLUMI Anti-NS1 is also conspicuously superior to the existing rapid diagnostic tests (RDT), with a sensitivity ranging from 48.5% to 80.7% [18,19,20,21]. Additionally, the MAGLUMI platform has developed the ultra-high-speed MAGLUMI X8, suitable for regions with a high prevalence of dengue fever, capable of meeting the demand of 600 tests per hour, and the small-sized MAGLUMI X3, suitable for working environments with constrained resources. The MAGLUMI X3 occupies an area of merely 0.68 m^2^ and has a speed of 200 tests per hour, jointly contributing to the establishment of a multi-level and three-dimensional dengue fever prevention and control system.

## Figures and Tables

**Figure 1 viruses-17-00106-f001:**
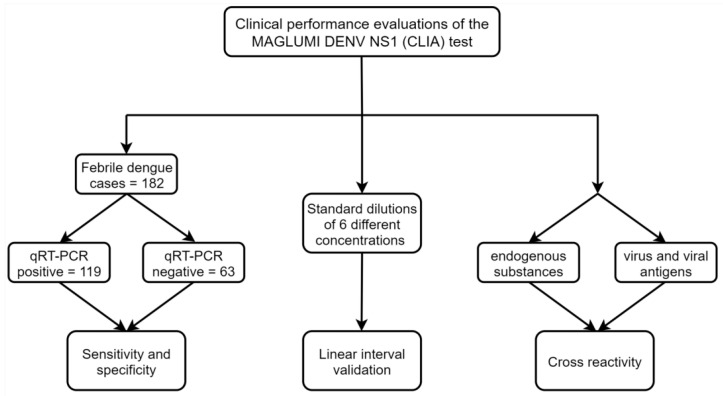
Study flow diagram. The MAGLUMI Denv NS1 test was validated for sensitivity and specificity using samples from 119 dengue confirmed cases and 63 dengue excluded cases. Linearity validation involved preparing samples with dilutions of NS1 antigen standard material. Cross-interference validation included samples containing endogenous interferents or potential cross-reacting substances.

**Figure 2 viruses-17-00106-f002:**
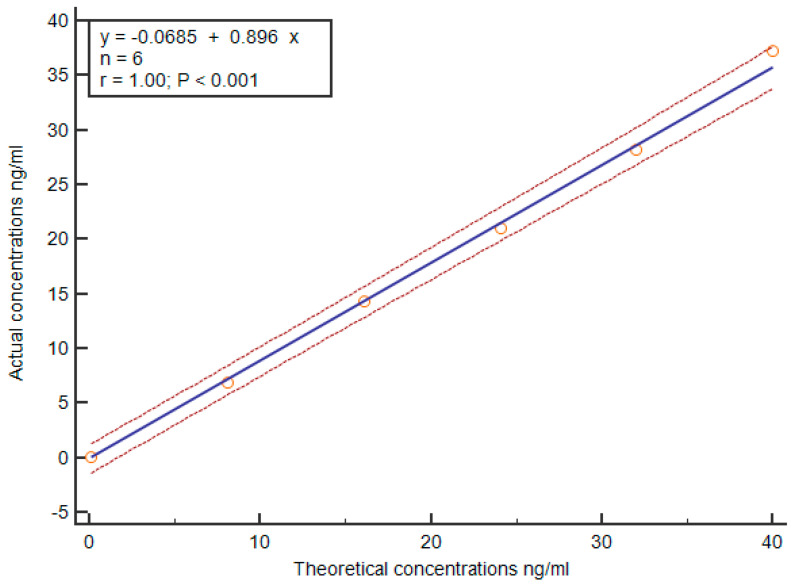
Reference intervals of the MAGLUMI Denv NS1 test. The tested concentrations of 6 samples (orange circles) were linearly fitted to the theoretical concentration through the least square method, generating a scatter diagram with regression line. Two curves on both sides were delineated to represent the 95% confidence interval.

**Table 1 viruses-17-00106-t001:** A 2 × 2 table reporting cross-classification of results generated by the MAGLUMI Denv NS1 test and the qPCR reference test.

MAGLUMI Denv NS1 Test	qPCR Reference Test		
	Positive	Negative	Total
Positive	117	1	118
Negative	2	62	64
Total	119	63	182

**Table 2 viruses-17-00106-t002:** Performance of the MAGLUMI Denv NS1 test on dengue-confirmed and dengue-excluded samples.

Measure	Calculation	Estimate	95% CI
Specificity	62/63	98.41%	91.54–99.72%
Sensitivity	117/119	98.32%	94.08–99.54%
FPR	1/63	1.59%	0.28–8.46%
FNR	2/119	1.68%	0.46–5.92%
PPV	117/118	99.15%	95.36–99.85%
NPV	62/64	96.88%	89.30–99.14%
Accuracy	179/182	98.35%	95.27–99.44%

FPR, false positive rate; FNR, false negative rate; PPV, positive predictive value; NPV, negative predictive value; CI, confidence interval.

**Table 3 viruses-17-00106-t003:** Sensitivity of the MAGLUMI Denv NS1 test in dengue qRT-PCR-positive samples.

	Serotype	# Samples	Reactive	Non-Reactive
Dengue qPCR positive samples	Unknown	4	3	1
	Serotype I	99	98	1
	Serotype II	14	14	0
	Serotype III	2	2	0
	Serotype IV	0	0	0
Total		119	117	2
Sensitivity			98.32%	
95% CI			94.08–99.54%	

**Table 4 viruses-17-00106-t004:** Analytical specificity of MAGLUMI Denv NS1 test in samples with potentially cross reacting agents and interfering substances.

		Reactive	Non-Reactive
Cross reacting substances/agents	Autoimmune diseases	0	6
	HAMA	0	3
	Rheumatoid factor positive	0	6
	TBE virus	0	4
	Yellow fever virus	0	3
	Japanese encephalitis virus	0	4
	West Nile virus	0	3
	Zika virus	0	3
	Zika NS1 recombinant antigen	0	2
	Chikungunya virus	0	4
	Measles virus	0	4
	Rubella virus	0	4
	Scarlet fever virus	0	4
	Leptospirosis	0	3
	HAV recombinant antigen	0	3
	HBsAg positive	0	5
	HCV recombinant antigen	0	3
Interfering substances	Acetaminophen	0	3
	Ibuprofen	0	3
	Aspirin	0	3
	Dexamethasone	0	3
	Hemolytic Low	0	2
	Hemolytic Medium	0	2
	Hemolytic High	0	2
	Lipemic Low	0	2
	Lipemic Medium	0	1
	Lipemic High	0	1
	Bilirubin	0	4
Total		0	90
analytical specificity			100%
95% CI			97.87–100%

HAMA, human anti-mouse antibody; HBsAg, hepatitis B surface antigen; HAV, hepatitis A virus; HCV, hepatitis C virus. Additional nine Denv NS1 recombinant antigen samples tested reactive, serving as reference tests for cross-interference analyzing.

**Table 5 viruses-17-00106-t005:** Analytic sensitivity for MAGLUMI Denv NS1 test.

Lot ID	Lot 1	Lot 2	Lot 3
LoB (AU/mL)	0.471	0.480	0.482
LoD (AU/mL)	0.725	0.702	0.743
LoQ (AU/mL)	0.863	0.917	0.914

**Table 6 viruses-17-00106-t006:** MAGLUMI Denv NS1 test results were obtained on NS1-negative or -positive serum/plasma samples.

Sample Type		Negative Samples		Positive Samples	
		Reactive	Non-Reactive	Reactive	Non-Reactive
Serum	/	0	8	11	0
	Glass powder	0	8	11	0
	Maleic acid and α-olefin	0	8	11	0
Plasma	Sodium citrate	0	8	11	0
	K_2_EDTA	0	8	11	0
	Lithium heparin	0	8	11	0
	Sodium heparin	0	8	11	0

EDTA, ethylenediaminetetraacetic acid; glass powder, including SiO_2_, Al_2_O_3_, and TiO_2_.

**Table 7 viruses-17-00106-t007:** Consistency validation of dengue virus NS1, IgG, and IgM diagnostic assays between the MAGLUMI and the EUROIMMUN.

Sample Size		EUROIMMUN Denv NS1				EUROIMMUN Denv IgG				EUROIMMUN Denv IgM	
114		+	−	195		+	−	134		+	−
MAGLUMI Denv NS1	+	6	1	MAGLUMI Denv IgG	+	77	2	MAGLUMI Denv IgM	+	21	1
	−	1	106		−	2	116		−	0	112
Sensitivity		85.71%				100.00%				100.00%	
Specificity		99.07%				98.31%				99.12%	
Accuracy		98.25%				98.97%				99.25%	

## Data Availability

Data is contained within the article or Appendix A. The original contributions presented in the study are included in the article/Appendix A, further inquiries can be directed to the corresponding authors.

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
