# Peer review of "Clinical Performance of MAGLUMI Diagnostic Tests for the Automated Detection of Dengue Virus"

_viruses, 2025, doi:10.3390/v17010106_

Round 1

Reviewer 1 Report

Comments and Suggestions for Authors

In this study, Peng B et al. evaluate the clinical performance of the MAGLUMI Denv NS1 antigen test for detecting dengue acute infection. Authors report that this chemiluminescent immunoassay utilizes dual antibodies to quantitatively detect NS1 antibodies in human serum/plasma. The manuscript reports a high specificity and sensitivity of >98% using 182 serum PCR-positive (n=119) or -negative specimens (n=63). The manuscript is well-written, and the presentations are clear. However, several areas lack clarity or miss vital information and must be addressed before publication. 

The authors' abbreviation of CLIA to refer to chemiluminescent immunoassay appears ambiguous compared to the well-recognized usage of CLIA for the Clinical Laboratory Improvement Amendments. Consider using a different abbreviation or choose not to abbreviate it. 

In lines 39-40, "Although the genomes of these five serotypes are similar" should be replaced with "Although the genomes of these four serotypes are similar."

In lines 63-64, the authors state, "Furthermore, the presence of NS1 antigen for up to 9 days or even longer can indicate active viral replication and potential severity of the disease." I suggest that the authors include a reference for this statement.

Since linearity verification data does not provide details on analytical sensitivity (limit of detection), I suggest adding this critical data to this paper. 

In lines 78-79, the authors claim that "we have developed a MAGLUMI Denv NS1 chemiluminescent immunoassay 78 (CLIA)." Their NS1 detection utilizes two monoclonal Abs to sandwich the NS1 antigen in plasma/serum. However, there are little details about the source of these monoclonal antibodies. Authors must include additional information about the monoclonal antibodies for transparency and reproducibility. 

In lines 200-207, the authors compare the Maglumi DENV IgG and IgM tests to Euroimmune DENV IgG and IgM tests. However, the manuscript method does not provide details about the Maglumi DENV IgG and IgM tests. I suggest the authors include additional information about them in the methods section. 

The authors discuss the lack of details on the performance of the NS1 detection against serotypes 3 and 4 but fail to address the performance of the test for detecting NS1 antigen in secondary dengue cases. This is crucial as Ab developed from the previous dengue infection(s) can lower the test's performance during secondary dengue infection by competing with the monoclonal antibodies used in the test. I suggest the authors perform the DENV IgG test with the acute specimen and report the NS1 Ag test performance in DENV IgG-positive (secondary dengue) and DENV IgG-negative (primary dengue) specimens.

Reviewer 2 Report

Comments and Suggestions for Authors

Bo Peng et al. have presented a study titled Clinical Performance of MAGLUMI Diagnostic Tests for the Automated Detection of Dengue Virus, evaluating the performance of MAGLUMI DENV NS1 (CLIA) in a retrospective analysis using residual samples validated by qPCR. The study reports an acceptable level of sensitivity and specificity, and the findings are promising.

 I have the following comments and suggestions for improvement:

1. The retrospective design of the study, coupled with a limited sample size, raises concerns about potential biases. Could the authors consider conducting prospective studies or field validations to strengthen the reliability of the findings?

2. The study includes only two samples from dengue serotype 3 and none from serotype 4, limiting the findings' generalizability to all dengue serotypes.

3. The MS lacks essential details regarding the residual samples used. Information about the sample size, geographic origin, and clinical characteristics would provide a clearer picture of the study's applicability and relevance to real-world settings.

4. I understand that cross-reactivity with Zika, chikungunya, and some other infectious diseases has been tested. However, consideration in evaluating the Japanese encephalitis (JE) virus and other flaviviruses with shared epitopes is important. 

5. While the study highlights the diagnostic utility of MAGLUMI DENV NS1 (CLIA), how does it compare with other assays of similar sensitivity and specificity? Additionally, how do the authors propose its adoption in resource-limited settings or outbreaks, given the availability of comparable alternatives?

Comments on the Quality of English Language

Minor language editing is suggested. 

Round 2

Reviewer 1 Report

Comments and Suggestions for Authors

The revised version has addressed my comments, and there are no further comments to add. Thank you.

Author Response

We are extremely grateful for your constructive comments and for the time and effort you have dedicated to reviewing our work. Thank you!

Reviewer 2 Report

Comments and Suggestions for Authors

Thank you to the authors for revising the manuscript "Clinical Performance of MAGLUMI Diagnostic Tests for the Automated Detection of Dengue Virus" and addressing most of my concerns. However, I recommend that the authors consider planning future studies, if not feasible at present, using a larger sample size, particularly for DENV 3 and DENV 4, before endorsing its use in dengue-endemic regions.  

Author Response

Comments 1: Thank you to the authors for revising the manuscript "Clinical Performance of MAGLUMI Diagnostic Tests for the Automated Detection of Dengue Virus" and addressing most of my concerns. However, I recommend that the authors consider planning future studies, if not feasible at present, using a larger sample size, particularly for DENV 3 and DENV 4, before endorsing its use in dengue-endemic regions.

Response 1: We are extremely grateful for your constructive comments and for the time and effort you have dedicated to reviewing our work. We have formulated a plan to carry out a clinical performance evaluation test for MAGLUMI Denv NS1 in the near future. The objective is to test 300 dengue fever samples in Brazil, Southeast Asia and southern China, including DENV 3 and DENV 4, in accordance with the standards of EP9-A2 and EP9-A3, to guarantee that all four serotypes can be detected by MAGLUMI Denv NS1.

"Line262-265: Given that all four serotypes of dengue virus spread in a similar manner, prior to the application of MAGLUMI Denv NS1 in dengue-endemic areas, verification will be carried out on a larger sample size, especially for samples of DENV 3 and DENV 4."